# Distinct mechanisms regulating mechanical force-induced Ca²⁺ signals at the plasma membrane and the ER in human MSCs

Tae-Jin Kim[1,2], Chirlmin Joo[3,4], Jihye Seong[1,5], Reza Vafabakhsh[3], Elliot L Botvinick[6], Michael W Berns[6], Amy E Palmer[7], Ning Wang[8], Taekjip Ha[3,9,10,11], Eric Jakobsson[2,12], Jie Sun[2,12]*, Yingxiao Wang[1,2,9,12,13,14]*

[1]Neuroscience Program, University of Illinois, Urbana-Champaign, Urbana, United States; [2]Beckman Institute for Advanced Science and Technology, University of Illinois, Urbana-Champaign, Urbana, United States; [3]Department of Physics, University of Illinois, Urbana-Champaign, Urbana, United States; [4]Kavli Institute of NanoScience and Department of BioNanoScience, Delft University of Technology, Delft, Netherlands; [5]Center for Neuro-Medicine, Korea Institute of Science and Technology, Seoul, Republic of Korea; [6]Department of Biomedical Engineering, Beckman Laser Institute, University of California, Irvine, Irvine, United States; [7]Department of Chemistry and Biochemistry, University of Colorado, Boulder, Boulder, United States; [8]Department of Mechanical Science and Engineering, University of Illinois, Urbana-Champaign, Urbana, United States; [9]Center for Biophysics and Computational Biology, University of Illinois, Urbana-Champaign, Urbana, United States; [10]Institute of Genomic Biology, University of Illinois, Urbana-Champaign, Urbana, United States; [11]Howard Hughes Medical Institute, University of Illinois, Urbana-Champaign, Urbana, United States; [12]Department of Molecular and Integrative Physiology, University of Illinois, Urbana-Champaign, Urbana, United States; [13]Department of Bioengineering, University of Illinois, Urbana-Champaign, Urbana, United States; [14]Department of Bioengineering, Institute of Engineering in Medicine, University of California, San Diego, San Diego, United States

*For correspondence: jiesun2@ illinois.edu (JS); yiw015@eng. ucsd.edu (YW)

**Abstract** It is unclear that how subcellular organelles respond to external mechanical stimuli. Here, we investigated the molecular mechanisms by which mechanical force regulates Ca²⁺ signaling at endoplasmic reticulum (ER) in human mesenchymal stem cells. Without extracellular Ca²⁺, ER Ca²⁺ release is the source of intracellular Ca²⁺ oscillations induced by laser-tweezer-traction at the plasma membrane, providing a model to study how mechanical stimuli can be transmitted deep inside the cell body. This ER Ca²⁺ release upon mechanical stimulation is mediated not only by the mechanical support of cytoskeleton and actomyosin contractility, but also by mechanosensitive Ca²⁺ permeable channels on the plasma membrane, specifically TRPM7. However, Ca²⁺ influx at the plasma membrane via mechanosensitive Ca²⁺ permeable channels is only mediated by the passive cytoskeletal structure but not active actomyosin contractility. Thus, active actomyosin contractility is essential for the response of ER to the external mechanical stimuli, distinct from the mechanical regulation at the plasma membrane.

## Introduction

Mechanical factors are known to play crucial roles in both development and tissue regeneration from stem cells. However, it remains unclear how these factors, such as mechanical forces, are converted

**eLife digest** Cells receive many signals from their environment, for example, when they are compressed or pulled about by neighboring cells. Information about these 'mechanical stimuli' can be transmitted within the cell to trigger changes in gene expression and cell behavior.

When a cell receives a mechanical stimulus, it can activate the release of calcium ions from storage compartments within the cell, including from a compartment called the endoplasmic reticulum. Calcium ions can also enter the cell from outside via channels located in the membrane that surrounds the cell (the plasma membrane).

Kim et al. investigated how mechanical forces are transmitted in a type of human cell called mesenchymal stem cells using optical tweezers to apply a gentle force to the outside of a cell. These tweezers use a laser to attract tiny objects, in this case a bead attached to proteins in the cell's outer membrane. The cell's response to this mechanical stimulation was measured using a sensor protein that fluoresces a different color when it binds to calcium ions.

With this set-up, Kim et al. found that mesenchymal stem cells are able to transmit mechanical forces to different depths within the cell. The forces can travel deep to trigger the release of calcium ions from the endoplasmic reticulum. This process involves a network of protein fibers that criss-cross to support the structure of a cell—called the cytoskeleton—and also requires proteins that are associated with the cytoskeleton to contract. However, calcium ion entry through the plasma membrane due to a mechanical force does not require these contractile proteins—only the cytoskeleton is involved.

These results demonstrate that the transmission of mechanical signals to different depths within mesenchymal stem cells involves different components. Future work should shed light on how these mechanical signals control gene expression and the development of mesenchymal stem cells.

into biochemical signals in stem cells to regulate regeneration processes. Calcium ion ($Ca^{2+}$) as one of the most important biochemical signals, is involved in many cellular processes, including muscle contraction, differentiation, proliferation, gene expression, and apoptosis (*Berridge et al., 2000*; *West et al., 2001*; *McKinsey et al., 2002*; *Landsberg and Yuan, 2004*; *Clapham, 2007*; *Maeda et al., 2007*; *Rong and Distelhorst, 2008*). Various mechanical stimulations can affect cytosolic $Ca^{2+}$ signals as well as $Ca^{2+}$ dynamics in organelles or subcellular compartments, such as mitochondria and focal adhesion sites (*Balasubramanian et al., 2007*; *Belmonte and Morad, 2008*; *Hayakawa et al., 2008*; *Horner and Wolfner, 2008*). To apply mechanical force precisely, we utilize optical laser tweezers to generate force in a bead coupled to the cell surface through the ligation of adhesion receptors and transmit it into the cell at subcellular locations to trigger signal transduction (*Berns, 2007*; *Botvinick and Wang, 2007*).

Most cells mobilize their $Ca^{2+}$ signals via the $Ca^{2+}$ entry across the plasma membrane and/or the $Ca^{2+}$ release from intracellular stores such as endoplasmic reticulum (ER) or sarcoplasmic reticulum (SR) (*Wehrens et al., 2005*; *Clapham, 2007*). Biochemical signals, such as inositol 1,4,5-trisphosphate ($IP_3$), are known to regulate $Ca^{2+}$ release from ER. But direct regulation of ER $Ca^{2+}$ signals by mechanical force is unknown. The human mesenchymal stem cells (HMSCs) displaying $Ca^{2+}$ oscillations provide a model system to study that (*Kawano et al., 2002*; *Sun et al., 2007*). As $Ca^{2+}$ influx at the plasma membrane and release from ER are the only two sources for $Ca^{2+}$ oscillations in HMSCs (*Kawano et al., 2002*; *Kim et al., 2009*), we dissected the effect of mechanical force on each process by monitoring the calcium signals at subcellular locations. To visualize $Ca^{2+}$ signal with high spatiotemporal resolutions, we employed a fluorescence resonance energy transfer (FRET)-based $Ca^{2+}$ biosensor and its variants anchored at subcellular organelles (*Miyawaki et al., 1997*; *Ouyang et al., 2008*). Here, we combined optical laser tweezers to deliver local mechanical force and FRET biosensors to investigate how mechanical force regulates $Ca^{2+}$ signals at different subcellular locations in HMSCs.

## Results and discussion

We first investigated how force regulates $Ca^{2+}$ release from ER with the FRET-based $Ca^{2+}$ biosensor (*Figure 1A*) to dissect the effect of mechanical force on each of the two $Ca^{2+}$ mobilization processes (*Figure 1B*). The experiments were done in the absence of extracellular $Ca^{2+}$ so that there was no $Ca^{2+}$

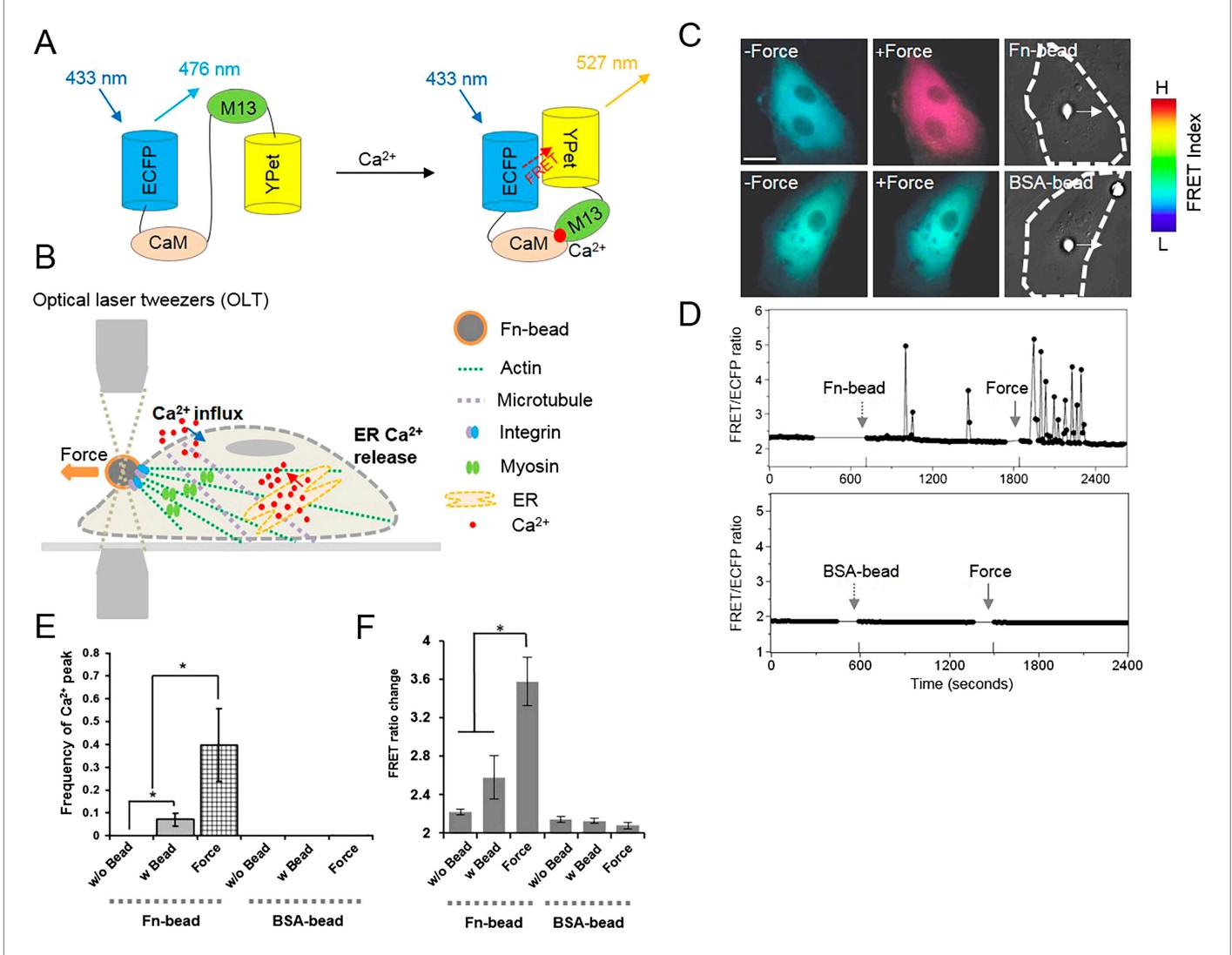

**Figure 1**. Intracellular $Ca^{2+}$ oscillations in response to mechanical force in HMSCs with $Ca^{2+}$-free medium. (**A**) A schematic drawing of the activation mechanism of the $Ca^{2+}$ FRET biosensor. (**B**) Beads coated with Fn or BSA were seeded onto the cell and mechanical force was applied by pulling a Fn-coated bead using optical laser tweezers. Both $Ca^{2+}$ influx and ER $Ca^{2+}$ release can contribute to force-induced $Ca^{2+}$ signals. (**C**) Color images represent the YPet/ECFP emission ratio of the cytoplasmic $Ca^{2+}$ biosensor. The color scale bars represent the range of emission ratio, with cold and hot colors indicating low and high levels of $Ca^{2+}$ concentration, respectively. (**D**) The time courses represent the YPet/ECFP emission ratio averaged over the cell body outside of nucleus upon seeding of Fn or BSA-coated beads and force application. (**E–F**) Bar graphs represent the frequency or ratio of the intracellular $Ca^{2+}$ oscillations evoked by mechanical force. Error bars indicate standard error of mean; *$p < 0.05$, n = 14. (Scale bar: 10 μm).

The following figure supplements are available for figure 1:

**Figure supplement 1**. Laser-tweezer pulling of a Fn-coated bead on a BAEC in $Ca^{2+}$-free medium.

**Figure supplement 2**. Mechanical force doesn't induce any increase in $IP_3$ level.

influx through plasma membrane and the oscillations were mainly from ER release (*Figure 1B*). When fibronectin (Fn)-coated beads were seeded onto the HMSCs and 300 pN of mechanical force was applied by optical laser tweezers as described previously (*Wang et al., 2005*), $Ca^{2+}$ oscillations were induced in HMSCs (*Figure 1C-F*, *Video 1*) but not in bovine aortic endothelial cells (BAECs) (*Figure 1—figure supplement 1*). In contrast, laser-tweezer-traction on bovine serum albumin (BSA)-coated beads

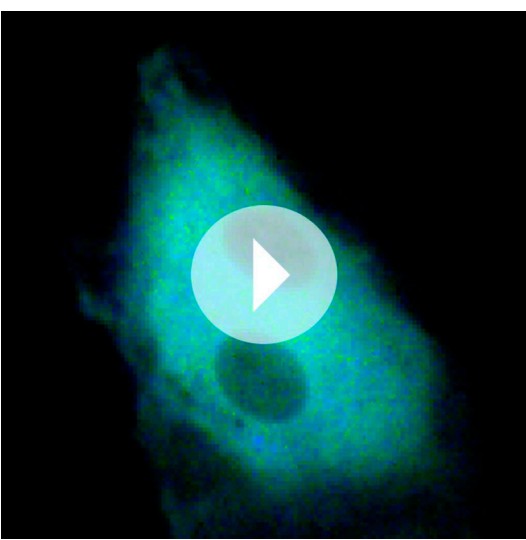

**Video 1**. A HMSC transfected with cytosolic Ca$^{2+}$ biosensors before and after mechanical force application by optical laser tweezers on a Fn-coated bead attached to the cell (Duration of Video: 2700 s). DOI: 10.7554/eLife.04876.006

did not cause any oscillations (*Figure 1C–F*). These results indicate that without extracellular Ca$^{2+}$, mechanical force can induce Ca$^{2+}$ oscillations by triggering Ca$^{2+}$ release from ER. Further experiments showed that depletion of ER Ca$^{2+}$ by Thapsigargin (TG) or inhibition of Ca$^{2+}$ release from the ER by 2-Amino-ethoxydiphenylborate (2-APB) entirely abolished the force-induced oscillations (*Figure 1—figure supplement 2A,B*). These results confirmed that the ER Ca$^{2+}$ store is the main source for the force-induced oscillations in HMSCs without extracellular Ca$^{2+}$.

Two possible mechanisms can regulate Ca$^{2+}$ release from ER upon mechanical laser-tweezer-traction. External force can either 1) transmit deep inside the cell and mechanically alter the channels on ER for Ca$^{2+}$ release (*Himmel et al., 1993*; *Missiaen et al., 1996*; *Rath et al., 2010*); or 2) trigger biochemical signaling cascades to produce IP$_3$ that diffuses inside to activate IP$_3$-sensitive Ca$^{2+}$ channels (*Lehtonen and Kinnunen, 1995*; *Matsumoto et al., 1995*). To distinguish them, we directly monitored the IP$_3$ level using a FRET-based biosensor, LIBRAvIIs. Mechanical force did not induce any change of IP$_3$ while ATP increased IP$_3$ production in HMSCs (*Figure 1—figure supplement 2C*), suggesting that the second mechanism is unlikely. Therefore, laser-tweezer-traction should transmit deep inside the cells to mechanically release Ca$^{2+}$ from ER.

Cytoskeleton is known to transmit mechanical forces and conduct mechanotransduction (*Hamill and Martinac, 2001*; *Orr et al., 2006*; *Schwartz and DeSimone, 2008*), so we investigated the role of cytoskeleton and its associated proteins in the force-induced ER Ca$^{2+}$ release. The disruption of cytoskeletal actin filaments by cytochalasin D (Cyto D) or microtubules by nocodazole (Noc) completely eliminated the force-induced Ca$^{2+}$ oscillations (*Figure 2A–B*). In addition, the inhibition of actomyosin contractility by ML-7 or blebbistatin had the same effect (*Figure 2C–D*). Thus, the deep penetration and transmission of force inside the cell and the induction of Ca$^{2+}$ release from ER depend on both cytoskeletal support and actomyosin contractility. This matches with the previous reports that long-distance force propagation to the deep cytoplasm depends on cytoskeleton tension (*Hu et al., 2003*) as well as the pivotal role of myosin light chain kinase (MLCK) and myosin II in regulating force development (*Olsson et al., 2004*; *Herring et al., 2006*; *Fajmut and Brumen, 2008*; *He et al., 2008*). There are two possible signals in ER that may trigger the Ca$^{2+}$ release in response to mechanical force. 1) IP$_3$R channels on the ER membrane are mechanosensitive and can be directly opened by transmitted mechanical force. Several lines of evidence supported this hypothesis. First, IP$_3$R is coupled to cytoskeleton and associated proteins allowing mechanical coupling. A direct binding between IP$_3$R and myosin II was discovered in *C. elegans* (*Walker et al., 2002*). In addition, IP$_3$R has linkage to actin mediated by an adaptor 4.1N protein (*Fukatsu et al., 2004*; *Turvey et al., 2005*). IP$_3$R also binds to ankyrins, which are adaptor proteins coupled to the spectrin-based cytoskeleton (*Bourguignon et al., 1993*; *Joseph and Samanta, 1993*). Second, IP$_3$R channel has an α-helix bundle at the pore forming region, similar to voltage-gated potassium and calcium channels (*Schug et al., 2008*), which are generally found to be mechanosensitive (*Morris, 2011*). The mechanism for their mechanosensitivity is possibly that the α-helix tilt angle tends to change when the membrane thins upon mechanical tension, in order to do proper hydrophobic matching with the interfacial region of the membrane, which leads to channel opening (*Cheng et al., 2004*; *Kim and Im, 2010*). Therefore, it is likely that the IP$_3$R channel is also mechanosensitive. 2) Other mechanosensitive channels on ER, for example, transient receptor potential (TRP) family, may also contribute to this force-induced Ca$^{2+}$ release. A number of TRP channels have been found to express at ER membranes, such as TRPC1 (*Berbey et al., 2009*), TRPV1 (*Gallego-Sandin et al., 2009*), TRPM8 (*Bidaux et al., 2007*), and TRPP2 (*Koulen et al., 2002*). As some TRP channels have been shown to be mechanosensitive and have linkage to cytoskeleton

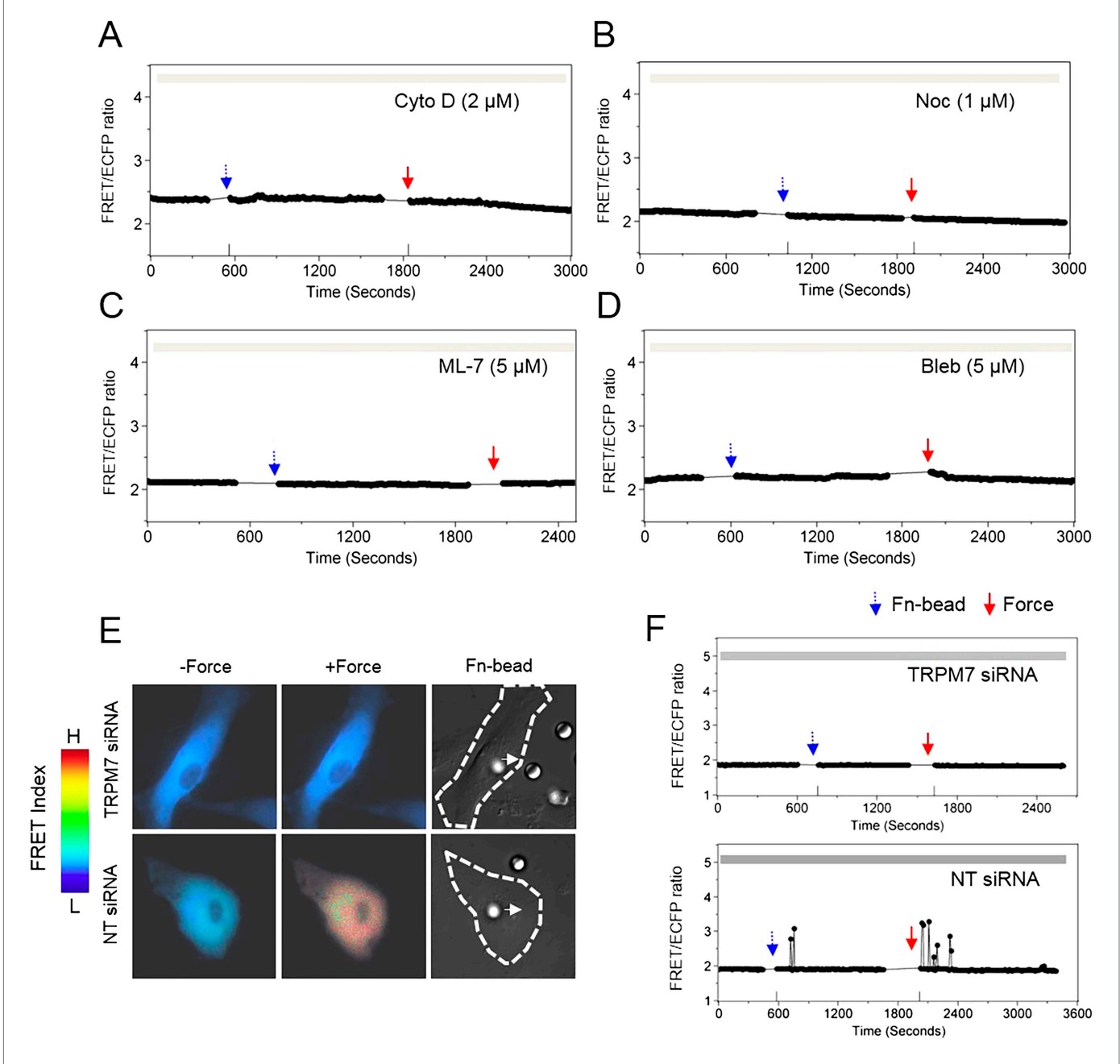

**Figure 2**. Cytoskeletal support, actomyosin contractility, and TRPM7 channels mediate the force-induced intracellular $Ca^{2+}$ oscillations. The time courses represent the YPet/ECFP emission ratio of cytoplasmic $Ca^{2+}$ in HMSCs in the absence of extracellular $Ca^{2+}$ when these cells were pretreated with (**A**) 2 μM Cyto D (n = 8), (**B**) 1 μM Noc (n = 8), (**C**) 5 μM ML-7 (n = 8), and (**D**) 5 μM Bleb (n = 8). (**E**) Color images represent the YPet/ECFP emission ratio of the cytoplasmic $Ca^{2+}$ biosensor in HMSCs transfected with NT or TRPM7 siRNA. The color scale bars represent the range of emission ratio, with cold and hot colors indicating low and high levels of $Ca^{2+}$ concentration, respectively. (**F**) The time courses represent the YPet/ECFP emission ratio averaged over the cell bodies outside of nucleus treated with siRNA as indicated.

The following figure supplements are available for figure 2:

**Figure supplement 1**. Stretch-activated or store-operated channels at the plasma membrane mediate force-induced $Ca^{2+}$ release from ER.

**Figure supplement 2**. TRPM7 channels as well as cytoskeletal support and actomyosin contractility mediate the force-induced intracellular $Ca^{2+}$ oscillations.

(*Barritt and Rychkov, 2005*), it is likely that TRP channels located at ER may mediate, at least in part, the force-induced ER calcium release. Notably, these two possibilities are not mutually exclusive as more than one type of channels can be responsible for the force-induced ER calcium release.

Surprisingly, blocking stretch-activated or store-operated channels at the plasma membrane by $Gd^{3+}$, $La^{3+}$, or streptomycin, but not L-type voltage-operated $Ca^{2+}$ channels by Nifedipine, abolished the mechanical force-induced $Ca^{2+}$ release from ER (*Figure 2—figure supplement 1*). These results suggest a possible coupling between force-induced $Ca^{2+}$ release at ER and stretch-activated and store-operated channels at the plasma membrane. As TRPM7 is one of the major $Ca^{2+}$ permeable mechanosensitive channels (*Wei et al., 2009*), we knocked down TRPM7 with targeting small interfering RNA (siRNA) to examine its role in the force-induced $Ca^{2+}$ oscillations. The decreased expression of TRPM7 and the lower percentile of HMSCs with $Ca^{2+}$ oscillations confirmed the effect of siRNA (*Figure 2—figure supplement 2A,B*) TRPM7 siRNA further abrogated the force-induced oscillations (*Figure 2E–F*). It is intriguing that the inhibition of TRPM7 at the plasma membrane can block the force transmission into ER in regulating calcium signals. Several possibilities may contribute to the observed results. 1) TRPM7 is functionally coupled to integrin, actomyocin contractility and cytoskeleton. As such, it may mediate and facilitate the force transmission to ER. Indeed, it has been shown that TRPM7 kinase can phosphorylate myosin II heavy chain (*Clark et al., 2006*) and regulate actomyocin contractility. 2) TRPM7 activity may have some downstream effect on $IP_3R$ in ER. For example, TRPM7 can control the protease calpain (*Su et al., 2006*), which can regulate $IP_3R$ degradation in ER (*Diaz and Bourguignon, 2000*). 3) TRPM7 may also be directly coupled to $IP_3R$ in the ER through adaptor proteins. Indeed, another TRP channel, TRPC1 has been shown to directly couple to $IP_3R$ in the ER through an adaptor protein Homer (*Yuan et al., 2003*). It becomes clear that membrane channels are not isolated entities floating in the plasma membrane. Instead, they are intimately coupled to integrins, cytoskeleton, actomyocin contractility, and ER membrane channels (*Cantiello et al., 2007*; *Matthews et al., 2007*; *Deng et al., 2009*). Therefore, these structural and physical couplings enable membrane channels to participate in direct force transmission to ER.

We further visualized the $Ca^{2+}$ release from ER directly, by generating and employing an improved ER-targeting $Ca^{2+}$ biosensor (D3ER) (*Palmer et al., 2006*; *Ouyang et al., 2008*). Mechanical force could directly induce the $Ca^{2+}$ release from ER without extracellular $Ca^{2+}$, as evidenced by the decrease in ER $Ca^{2+}$ concentration (*Figure 3A-B*, *Video 2*). The measurements of direct $Ca^{2+}$ release from ER in the presence of inhibitors or siRNA of TRMP7 were also consistent with our observations when cytosolic $Ca^{2+}$ biosensor was used (*Figure 3C*).

To gain more insights, we then examined the effect of force on $Ca^{2+}$ influx at the plasma membrane. ER $Ca^{2+}$ release was blocked by an $IP_3Rs$ inhibitor, 2-APB, in the presence of extracellular $Ca^{2+}$ so that the only source of intracellular $Ca^{2+}$ change is the calcium influx from extracellular space (*Bootman et al., 2002*). Nifedipine did not have any effect on the force-induced $Ca^{2+}$ influx, while $Gd^{3+}$, $La^{3+}$, streptomycin, and TRPM7 siRNA significantly inhibited this $Ca^{2+}$ influx (*Figure 4A–B*), confirming the involvement of mechanosensitive channels, especially TRPM7, which was shown to be directly activated by a membrane stretch or shear stress in various cell types (*Numata et al., 2007a*, *2007b*; *Wei et al., 2009*). The disruption of actin filaments or microtubules also inhibited the force-induced $Ca^{2+}$ influx (*Figure 4C–D*), suggesting that cytoskeletal integrity is essential for the membrane channels to respond to force, consistent with previous reports (*Hayakawa et al., 2008*). Interestingly, ML-7 or blebbistatin did not affect this force-induced $Ca^{2+}$ influx, suggesting that active actomyosin contractility may not be involved in the mechano-regulation of membrane channels (*Figure 4C–D*), different from their indispensable roles in the force-induced ER $Ca^{2+}$ release (*Figure 4E*).

Since mechanical stimulation such as stretch affects integrin adhesion and integrin-associated signaling, including Src and focal adhesion kinase (FAK) (*Wang et al., 2005*; *Orr et al., 2006*), we investigated whether Src and FAK mediate the force-induced $Ca^{2+}$ signals. Our results indicated that neither PP1, an inhibitor of Src family kinases, nor PF228, an inhibitor of FAK, had a significant effect on the force-induced $Ca^{2+}$ influx or ER release (*Figure 4—figure supplement 1*). Inhibition of phosphoinositide 3-kinases (PI3Ks) with LY294002 did not have any effect either (*Figure 4—figure supplement 1*). Although integrin-mediated Src phosphorylation and activation can regulate L-type $Ca^{2+}$ channel functions (*Gui et al., 2010*), the force-induced $Ca^{2+}$ oscillations were clearly independent of L-type $Ca^{2+}$ channels in HMSCs (*Figure 2—figure supplement 1*). Mechanical stretch can induce the phosphorylation of FAK and nitric oxide formation in cardiomyocytes (*Pinsky et al., 1997*; *Khan et al., 2003*), which can modulate ryanodine receptor 2 (RYR2) and $Ca^{2+}$ release at the

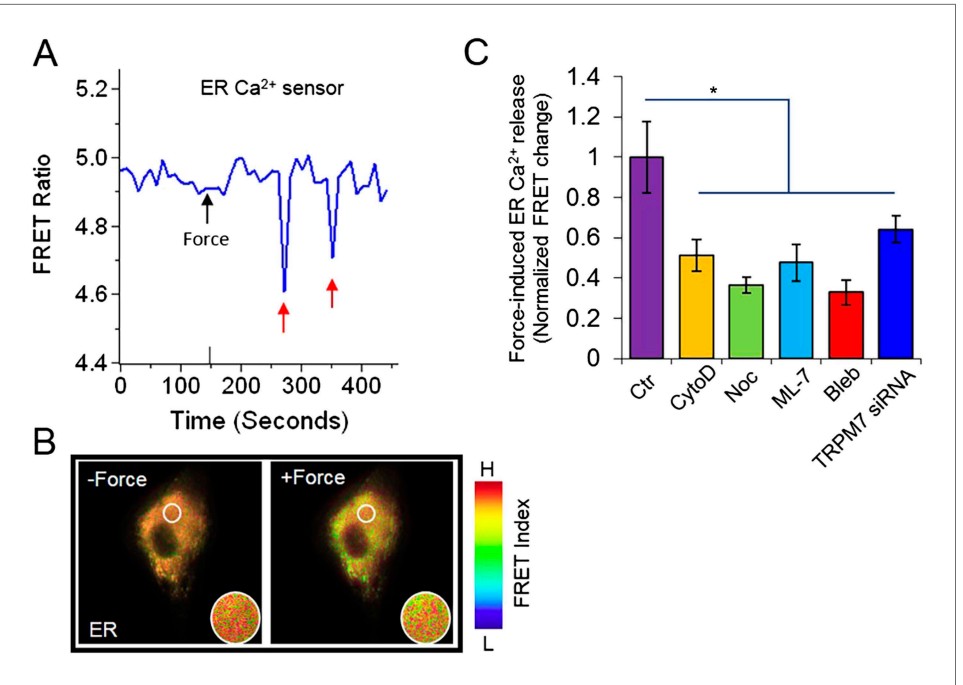

**Figure 3**. The visualization of force-induced $Ca^{2+}$ release from ER using a FRET-based ER $Ca^{2+}$ biosensor. (**A**) The time course and (**B**) the color images of YPet/ECFP emission ratio in HMSCs expressing the D3ER biosensor before and after force application. The red arrows indicated episodes of $Ca^{2+}$ release from ER. (**C**) The bar graphs represent the normalized changes of YPet/ECFP emission ratio of the D3ER in HMSCs upon force application without extracellular $Ca^{2+}$ in the untreated cells as the control group (n = 3) or cells pretreated with CytoD (n = 5), Noc (n = 5), ML-7 (n = 6), Bleb (n = 5), or TRPM7 siRNA (n = 9) as indicated. * represents $p < 0.05$.

SR (*Vila Petroff and Mattiazzi, 2001*). However, there is minimal expression of RYRs in HMSCs (*Kawano et al., 2002*), which may explain the lack of FAK involvement in the mechano-regulation of $Ca^{2+}$ signals. Together, the biochemical activities above have no detected involvement in the force-induced $Ca^{2+}$ signals.

Further analysis of the duration between force application and the first $Ca^{2+}$ signal indicated that in the absence of extracellular $Ca^{2+}$, the average delay time of ER calcium release monitored by both cytosolic $Ca^{2+}$ biosensor and ER $Ca^{2+}$ biosensor are around 100 s (*Figure 4—figure Supplement 1C*). Meanwhile, the average delay time of the calcium influx through plasma membrane in the presence of 2-APB and extracellular calcium is much shorter, around 20 s (*Figure 4—figure supplement 1C*), indicating faster response to mechanical force stimulation. Although the delay time of ER calcium release is longer than that of calcium influx across the plasma membrane, it does not contradict mechanical signal-mediated mechanism. Several reasons may contribute to longer time delay of ER calcium release. 1) The mechanical coupling machinery may need time for reinforcement to allow sufficient force transfer between the plasma membrane and the ER membrane. It is likely that mechanical force transmits through integrins and cytoskeleton (*Wang et al., 1993*; *Perozo and Rees, 2003*) as only Fn-coated, but not BSA-coated beads can induce ER calcium release (*Figure 1C–F*). However, the discrete network linkage of the existing cytoskeleton at the time of force application may not be sufficient for force focusing and transmission to ER, particularly because the apical integrins bound to the Fn-coated beads had not previously experienced force and therefore may have limited connection with cytoskeleton (*Matthews et al., 2006*). As force can unfold proteins, change their conformations to expose cryptic binding sites to allow the assembly of new/stronger network linkages (*del Rio et al., 2009*; *Johnson et al., 2007*), mechanical force can modulate the structural coupling of molecules and mechanical properties of the cells through cytoskeletal remodeling (*Kaazempur Mofrad et al., 2005*; *Matthews et al., 2006*; *Rosenblatt et al., 2007*). These modulation processes of mechanical coupling and reinforcement may need time to reach the threshold for sufficient mechanotransduction of ER calcium release; 2) the channels on ER membrane could have different kinetics

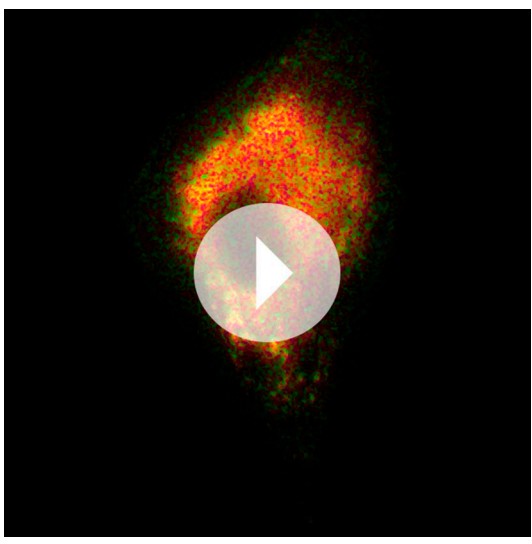

**Video 2**. A HMSC transfected with ER Ca²⁺ biosensors before and after mechanical force stimulation by optical laser tweezers on a Fn-coated bead attached to the cell (Duration of Video: 450 s).

and mechanosensitivity (*Moe et al., 1998*), therefore they may need larger focused and reinforced force to be developed at the site of ER to reach the threshold for physical opening. All these factors may contribute to the longer delay time for force-induced ER calcium release. Again our model does not rule out biochemical signal-mediated mechanisms. As a matter of fact, biochemical signal-mediated mechanisms, such as protein–protein interaction and cytoskeletal remodeling under mechanical tension are important to mediate mechanical signals as discussed above.

As such, our results have provided molecular insights on how mechanical force triggers intracellular Ca²⁺ oscillations through two mechanisms in HMSCs: Ca²⁺ influx at the plasma membrane and Ca²⁺ release from ER (*Figure 1B*). Our results showed that the deep penetration and transmission of mechanical force to regulate ER functions is dependent on not only the passive cytoskeletal support of actin filaments and microtubules, but also the active actomyosin contractility controlled by MLCK and myosin II. In contrast, the passive cytoskeletal support, but not active actomyosin contractility, is needed for the mechanotransduction at the plasma membrane levels, including the mechanoactivation of channels and Ca²⁺ influx across the plasma membrane (*Figure 4E*). These results hence provide direct evidence that the mechanotransduction at different depths of cell body is mediated by differential sets of mechanosensing elements.

## Materials and methods

### Gene construction and DNA plasmid

The construct of FRET-based Ca²⁺ biosensor has been described well in our previous articles (*Ouyang et al., 2008*; *Kim et al., 2009*). In brief, the fragment containing enhanced cyan fluorescent protein (ECFP), calmodulins (CaMs), and M13 was fused to YPet and subcloned into pcDNA3.1 (Invitrogen, Carlsbad, CA) for mammalian cell expression by using BamHI and EcoRI sites. The ECFP/YPet pair has allowed a higher sensitivity of FRET biosensors than those based on ECFP/Citrine pair. To generate an improved ER-targeting Ca²⁺ biosensor, the mutant peptide and CaMs regions were replaced with those regions of D3cpv and cloned between a truncated ECFP and YPet. For the ER targeting motifs, the calreticulin signal sequence MLLPVLLLGLLGAAAD was added 5′ to ECFP, and an ER retention sequence KDEL to the 3′ end of YPet. The construct of a FRET-based IP3 biosensor, LIBRAvIIs was kindly provided by Professor Akihiko Tanimura at University of Hokkaido, Japan (*Tanimura et al., 2009*).

### Cell culture and transfection

Human mesenchymal stem cells (HMSCs) and bovine aortic endothelial cells (BAECs) were obtained from the American Type Culture Collection (ATCC, Rockville, MD). HMSCs and BAECs were cultured in human mesenchymal stem cell growth medium (MSCGM, PT-3001, Lonza Walkersville, Inc., Walkersville, MD) and in Dulbecco's modified Eagle's medium (DMEM), respectively, supplemented with 10% fetal bovine serum (FBS), 2 mM L-glutamine, 100 U/ml penicillin, and 100 μg/ml streptomycin. The cells were cultured in a humidified incubator of 95% O₂ and 5% CO₂ at 37°C. The DNA plasmids were transfected into the cells by using Lipofectamine 2000 (Invitrogen, Carlsbad, CA) according to the product instructions.

### RNA interference assays

Double-stranded small interfering RNA (siRNA) sequences targeting human TRPM7 (ON-TARGETplus SMARTpool siRNA) and non-targeting control sequences were designed by Dharmacon RNAi

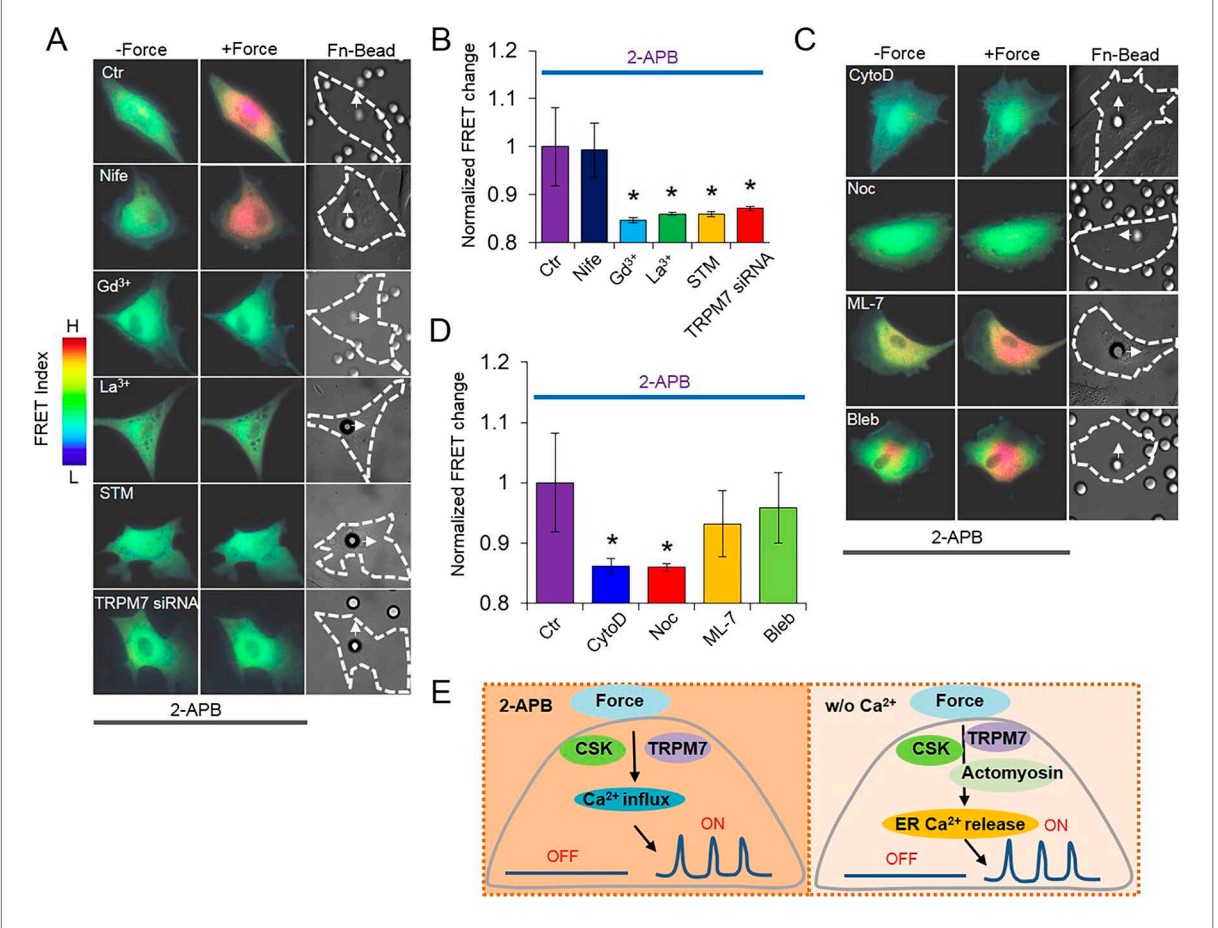

**Figure 4**. Ca²⁺ influx in response to mechanical force in Ca²⁺-containing medium. Ca²⁺ release from ER in all the HMSCs was blocked by pretreatment with 2-APB. (**A**, **C**) Color images represent the YPet/ECFP emission ratio of the cytoplasmic Ca²⁺ biosensor in control cells treated by 2-APB only (n = 5) or those co-treated by Nifedipine (n = 5), Gd³⁺ (n = 3), La³⁺ (n = 6), STM (n = 8), or TRPM7 siRNA (n = 9), CytoD (n = 8), Noc (n = 4), ML-7 (n = 9) or blebbistatin (n = 6). Arrows in DIC images point to the direction of applied force. (**B**, **D**) Bar graphs represent the normalized change of YPet/ECFP emission ratio of the cytoplasmic Ca²⁺ biosensor under different conditions as indicated in (**A**, **C**). Error bars indicate standard errors of mean; * represents $p < 0.05$. (**E**) The models depicting the mediators of mechanical force-induced Ca²⁺ influx or ER Ca²⁺ release.

The following figure supplement is available for figure 4:

**Figure supplement 1**. Src, FAK or PI3K has no effect on the mechanical force-induced Ca²⁺ signals.

Technology (Dharmacon Inc., Lafayette, CO). HMSCs were transfected with 1–2 µg siRNA specific for TRPM7 or a non-silencing control sequence according to the product instructions.

## Western blotting

The cells transfected with TRPM7 or non-targeting siRNA were washed twice with cold phosphate buffered saline (PBS) and then lysed in lysis buffer containing 50 mM Tris, pH 7.4, 150 mM NaCl, 1 mM EDTA, 1% Triton X-100, and a mix of serine and cysteine protease inhibitors. Lysates were centrifuged at 10,000×*g* at 4°C for 10 min. Cell lysates were then applied to 15% SDS-polyacrylamide gel electrophoresis, transferred to nitrocellulose, blocked with 5% non-fat milk, and detected by Western blotting using polyclonal goat anti-TRPM7 antibody (1:100; Abcam Inc., Cambridge, MA).

## Immunostaining

A polyclonal antibody against TRPM7 was used in both normal HMSCs and TRPM7-knockdown HMSCs. After being washed in cold phosphate buffered saline (PBS), the samples were fixed by

4% paraformaldehyde in PBS at room temperature for 15 min. The samples were incubated with a goat polyclonal antibody against TRPM7 (1:100; Abcam Inc., Cambridge, MA) at room temperature for 2 hr, followed by the incubation with TRITC-conjugated anti-goat IgG (1:100, Jackson ImmunoResearch Lab., Inc., West Grove, PA) at room temperature for 1 hr before the mounting of anti-photobleaching reagent (Vector Lab., Inc., Burlingame, CA).

## Solutions and chemicals

Imaging experiments were conducted with $Ca^{2+}$ free Hanks balanced salt solution (HBSS, Invitrogen) containing 20 mM HEPES, 1 mM D-glucose, 0.5 mM EGTA, 1 mM $MgCl_2$, and 1 mM $MgSO_4$ (pH 7.4). During imaging experiments, the solution was kept in streptomycin free condition to prevent a possible effect on mechanosensitive ion channels. The chemical reagents 2-Amino-ethoxydiphenyl borate (2-APB), Nifedipine, Thapsigargin (TG), $LaCl_3$, $GdCl_3$, streptomycin, nocodazole, cytochalasin D, blebbistatin, and ML-7 were purchased from Sigma–Aldrich (Sigma, St. Louis, MO). PP1 and LY294002 were commercially obtained from Calbiochem (San Diego, CA). PF228 was obtained from Tocris Bioscience (Ellisville, MO). The amount of drug administration was based on previous publications (*Kim et al., 2009*; *Lu et al., 2011*; *Seong et al., 2011*).

## Preparation of beads and optical laser tweezers

A fiber-coupled IR (infra-red) laser (1064 nm, 5W, 5 mm diameter, YLD-5-1064-LP, IPG Photonics) was used for the experiment. We used a piezoelectric system for a steering mirror and the piezo-mirror system was designed with a closed loop, and an automated shutter (LS6ZM2, Uniblitz, Rochester, NY) with a shutter controller (VCM-D1, Uniblitz). Mirrors (designed for IR), lenses (BK7, plano-convex), and other basic optics were purchased from Thorlabs (Newton, New Jersey). A hot mirror (FM01, wide band, Thorlabs) was installed inside a microscope to block the IR scattering. The piezo-mirror (S-334.2, (PI) Physik instrumente) was installed together with the computer interface module of (E-516.I3, PI). The interface module set up with the other drivers (E-503.00 and E-509.S3, PI) makes it possible to control the piezo-mirror system by a computer. The laser beam passes through a laser-beam expander, a steering mirror, and a dichroic long-pass beam splitter to enter the microscope side port. Beads coated with fibronectin (Fn; 50 µg/ml) or BSA as the control were prepared as previously reported (*Wang et al., 2006*). The size of the beads is 10 µm and the beads were incubated for 10–20 min to allow them to adhere to cell membrane surface. Single-beam gradient optical laser tweezers with controlled 300 pN of mechanical force were applied to pull the adhered beads. A similar optical trapping system has been described in our previous report (*Botvinick and Wang, 2007*).

## Microscopy, imaging acquisition, and analysis

Cells expressing various exogenous proteins were starved with 0.5% FBS for 36–48 hr before imaging experiments. All images were obtained by using Zeiss Axiovert inverted microscope equipped with a charge-coupled device (CCD) camera (Cascade 512B, Photometrics) and a 420DF20 excitation filter, a 450DRLP dichroic mirror, and two emission filters controlled by a filter changer (480DF30 for ECFP and 535DF25 for YPet). Time lapse fluorescence images were acquired at 10 s interval by MetaFluor 6.2 software (Universal Imaging, West Chester, PA). The emission ratio of YPet/ECFP was directly computed and generated by the MetaFluor software to represent the FRET efficiency before they were subjected to quantification and analysis by Excel (Microsoft, Redmond, WA).

## Statistical analysis

The results were expressed as the mean standard error of the mean (S.E.M). Statistical analysis of the data was performed by the unpaired Student's *t*-test to determine the statistical differences between the two mean values. The statistically significant level was determined by $p < 0.05$.

## Acknowledgements

We would like to thank Drs. Akihiko Tanimura and Xuefeng Wang for the generous gift of $IP_3$ reporters and valuable discussions, respectively. This work was supported by grants National Institute of Health (NIH) HL098472, HL109142, HL121365, GM106403, National Science Foundation (NSF) CBET0846429, CBET1344298, DMS1361421 (YW), and NSF 0822613, 0646550 (TJH), and NIH GM072744 (NW).

## Additional information

### Competing interests

TH: Reviewing editor, *eLife*. The other authors declare that no competing interests exist.

### Funding

| Funder | Grant reference number | Author |
|---|---|---|
| National Institutes of Health | HL098472, GM106403, NS063405 | Yingxiao Wang |
| National Science Foundation | CBET0846429, CBET1344298, DMS1361421 | Yingxiao Wang |
| National Institutes of Health | GM072744 | Ning Wang |
| National Science Foundation | 0822613, 0646550 | Taekjip Ha |

The funders had no role in study design, data collection and interpretation, or the decision to submit the work for publication.

### Author contributions

T-JK, Conception and design, Acquisition of data, Analysis and interpretation of data, Drafting or revising the article; CJ, Conception and design, Analysis and interpretation of data, Drafting or revising the article; JS, Conception and design, Acquisition of data, Analysis and interpretation of data; RV, Conception and design, Acquisition of data; ELB, MWB, Conception and design, Analysis and interpretation of data; AEP, Analysis and interpretation of data, Contributed unpublished essential data or reagents; NW, TH, Conception and design, Analysis and interpretation of data; EJ, Analysis and interpretation of data, Drafting or revising the article; JS, YW, Conception and design, Analysis and interpretation of data, Drafting or revising the article

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
