## [Decision Letter]

Thank you for sending your work entitled “Distinct mechanisms regulating mechanical force-induced Ca^2+^ signals at the plasma membrane and ER in human MSCs” for consideration at *eLife*. Your article has been favorably evaluated by Fiona Watt (Senior editor), a Reviewing editor, and 3 reviewers.

The Reviewing editor and the reviewers discussed their comments before we reached this decision, and the Reviewing editor has assembled the following comments to help you prepare a revised submission.

We find the results on the independent roles of ER calcium ion release and calcium ion channels in force-induced calcium transients very interesting, and important for advancing the field of mechanobiology. While it has been known that calcium release can be triggered by external forces applied to cells, this is the first time that evidence has been presented for when one or the other mechanism is activated following the application of force. Also, the ER-specific calcium sensor used in the absence of extracellular calcium provided an elegant approach for identifying the role of ER calcium channels. We believe that this approach will be useful in future detailed studies of force-induced calcium release.

We invite the authors to address the following criticism raised by the reviewers:

1) The IP3 is elevated in responding to the mechanic force by optical laser tweezer-mediated bead traction and so it is not responsible for Ca^2+^ release from ER. The authors concluded that the force was deeply penetrated to the ER to trigger Ca^2+^ release. This is a very interesting discovery. However, the authors should provide some clues about what kind of signals in ER that may trigger the Ca^2+^ release in responding to mechanic force.

2) siRNA for TRPM7 almost completely abolished the Ca^2+^ release (Figure 2), indicating that it may play an essential role in mechanic force-induced Ca^2+^ release. The authors should explain how it is coupled to the Ca^2+^ in ER, because both pathways contribute to the Ca^2+^ flux.

3) If two pathways; ER and plasma membrane, contribute to the Ca^2+^ flux in responding to the mechanic force, please explain why the inhibition by Gd^3+^, La^3+^, or streptomycin, blocked all the Ca^2+^ signal (Figure 2—figure supplement 1)?

4) The experiments demonstrate that the force transmission to the ER is an important prerequisite for ER calcium release. The finding that this effect can be blocked by inhibiting the plasma membrane channels is both intriguing and worrisome. It is hard to imagine how blocking the plasma membrane channels would affect force transmission to the ER, raising the prospect that there is some other form of signal transmission that causes plasma membrane channel activation to produce activation of the ER. If so, then this would counter all of the arguments supporting the role of direct force transmission to the ER as the primary mechanism for calcium release there. This is a point that deserves additional discussion.

5) No data are presented regarding the delay time between force application and the first ensuing calcium transient. One might expect the channels to respond as soon as force is transmitted, which should be virtually instantaneous. Alternatively, a longer delay might be indicative of a biochemical signal mediated mechanism. This also deserves discussion.

---

## [Author Response]

*1) The IP3 is elevated in responding to the mechanic force by optical laser tweezer-mediated bead traction and so it is not responsible for Ca*^*2+*^
*release from ER. The authors concluded that the force was deeply penetrated to the ER to trigger Ca*^*2+*^
*release. This is a very interesting discovery. However, the authors should provide some clues about what kind of signals in ER that may trigger the Ca*^*2+*^
*release in responding to mechanic force*.

We thank the reviewers for this important comment. There are two possible signals in ER that may trigger the Ca^2+^ release in response to mechanical force. 1) IP_3_R channels on the ER membrane are mechanosensitive and can be directly opened by transmitted mechanical force. Several lines of evidence supported this hypothesis. First, IP_3_R is coupled to cytoskeleton and associated proteins allowing mechanical coupling. A direct binding between IP_3_R and myosin II was discovered in *C. elegans* (68). In addition, IP_3_R has linkage to actin mediated by an adaptor 4.1N protein (19; 66). IP_3_R also binds to ankyrins, which are adaptor proteins coupled to the spectrin-based cytoskeleton (10; 30). Second, IP_3_R channel has an α-helix bundle at the pore forming region, similar to voltage-gated potassium and calcium channels (60), which are generally found to be mechanosensitive (48). The mechanism for their mechanosensitivity is possibly that the α-helix tilt angle tends to change when the membrane thins upon mechanical tension, in order to do proper hydrophobic matching with the interfacial region of the membrane, which leads to channel opening (12; 34). Therefore it is likely that the IP_3_R channel is also mechanosensitive. 2) Other mechanosensitive channels on ER, e.g. TRP family, may also contribute to this forced-induced Ca^2+^ release. A number of TRP channels have been found to express at ER membranes, such as TRPC1 (4), TRPV1 (20), TRPM8 (7), TRPP2 (Polycystin-2) (36). As some TRP channels have been shown to be mechanosensitive and have linkage to cytoskeleton (2), it is likely that TRP channels located at ER may mediate, at least in part, the force-induced ER calcium release. Notably, these two possibilities are not mutually exclusive as more than one type of channels can be responsible for the force-induced ER calcium release. The discussion above has been incorporated in the manuscript (please see the third paragraph of the Results and Discussion section).

*2) siRNA for TRPM7 almost completely abolished the Ca*^*2+*^
*release (*Figure 2*), indicating that it may play an essential role in mechanic force-induced Ca*^*2+*^
*release. The authors should explain how it is coupled to the Ca*^*2+*^
*in ER, because both pathways contribute to the Ca*^*2+*^
*flux*.

Several possibilities may contribute to the observed results. 1) TRPM7 is functionally coupled to integrin, actomyocin contractility and cytoskeleton. As such, it may mediate and facilitate the force transmission to ER. Indeed, it has been shown that TRPM7 kinase can phosphorylate myosin II heavy chain (14) and regulate actomyocin contractility. 2) TRPM7 activity may have some downstream effect on IP_3_R in ER. For example, TRPM7 can control the protease calpain (63), which can regulate IP_3_R degradation in ER (17). 3) TRPM7 may be directly coupled to IP_3_R in the ER through adaptor proteins. Indeed, another TRP channel, TRPC1 has been shown to directly couple to IP_3_R in the ER through an adaptor protein Homer (75). The discussion above has been incorporated in the manuscript (please see the fourth paragraph of the Results and Discussion section).

*3) If two pathways; ER and plasma membrane, contribute to the Ca*^*2+*^
*flux in responding to the mechanic force, please explain why the inhibition by Gd*^*3+*^*, La*^*3+*^*, or streptomycin, blocked all the Ca*^*2+*^
*signal (*Figure 2—figure supplement 1*)?*

We performed Gd^3+^, La^3+^, and streptomycin inhibition experiments first. As they are board inhibitors for stretch-activated channels or store-operated calcium channels, we concluded that mechanosensitive channels are generally essential for the force-induced calcium release. We then conducted the TRPM7 siRNA experiment to reveal that TRPM7 as a potential mechanosensitive channel candidate is essential for the force-induced calcium release. Accordingly, this question is essentially similar to question #2: how are mechanosensitive channels at the plasma membrane coupled to ER calcium release, which has been addressed in responses to this question as shown above. In general, it is hypothesized that mechanosensitive channels at the plasma membrane can potentially mediate the force-induced calcium release at ER by modulating force transmission efficiency, mechanically altering IP_3_R at ER through physical coupling, as well as affecting IP_3_R functions at ER. It is of note that other mechanosensitive channels, including other TRP family channels besides TRPM7, may also be involved in this force transmission process.

*4) The experiments demonstrate that the force transmission to the ER is an important prerequisite for ER calcium release. The finding that this effect can be blocked by inhibiting the plasma membrane channels is both intriguing and worrisome. It is hard to imagine how blocking the plasma membrane channels would affect force transmission to the ER, raising the prospect that there is some other form of signal transmission that causes plasma membrane channel activation to produce activation of the ER. If so, then this would counter all of the arguments supporting the role of direct force transmission to the ER as the primary mechanism for calcium release there. This is a point that deserves additional discussion*.

We want to thank the reviewers for the question. This question is similar to questions #2 and #3, i.e. how can membrane channels affect force transmission to the ER. Accordingly, please see our responses to these questions above. Indeed, it becomes clear that membrane channels are not isolated entities floating in the plasma membrane. Instead, they are intimately coupled to integrins, cytoskeleton, actomyocin contractility and ER membrane channels (11; 16; 43). Therefore, these structural and physical couplings enable membrane channels to participate in direct force transmission to ER. The discussion above has been incorporated in the manuscript (please see the fourth paragraph of the Results and Discussion section).

*5) No data are presented regarding the delay time between force application and the first ensuing calcium transient. One might expect the channels to respond as soon as force is transmitted, which should be virtually instantaneous. Alternatively, a longer delay might be indicative of a biochemical signal mediated mechanism. This also deserves discussion*.

We want to thank the reviewers for this inspiring comment. We have compiled the data of delay time in each of the three groups and added the graph as Figure 4—figure supplement 1. (Figure caption has been added.) Our data indicated that in the absence of extracellular Ca^2+^, the average delay time of ER calcium release monitored by both cytosolic Ca^2+^ biosensor and ER Ca^2+^ biosensor are around 100 sec. Meanwhile, the average delay time of the calcium influx through plasma membrane in the presence of 2APB and extracellular calcium is much shorter, around 20 sec, indicating faster response to mechanical force stimulation.

Although the delay time of ER calcium release is longer than that of calcium influx across the plasma membrane, it does not contradict mechanical signal-mediated mechanism. Several reasons may contribute to longer time delay of ER calcium release. 1) The mechanical coupling machinery may need time for reinforcement to allow sufficient force transfer between the plasma membrane and the ER membrane. It is likely that mechanical force transmits through integrins and cytoskeleton (55; 69) as only Fn-coated, but not BSA-coated beads can induce ER calcium release (Figure 1). However, the discrete network linkage of the existing cytoskeleton at the time of force application may not be sufficient for force focusing and transmission to ER, particularly because the apical integrins bound to the Fn-coated beads had not previously experienced force and therefore may have limited connection with cytoskeleton (42). As force can unfold proteins, change their conformations to expose cryptic binding sites to allow the assembly of new/stronger network linkages (15; 29), mechanical force can modulate the structural coupling of molecules and mechanical properties of the cells through cytoskeletal remodeling (31; 42; 59). These modulation processes of mechanical coupling and reinforcement may need time to reach the threshold for sufficient mechanotransduction of ER calcium release; 2) the channels on ER membrane could have different kinetics and mechanosensitivity (47), therefore they may need larger focused and reinforced force to be developed at the site of ER to reach the threshold for physical opening. All these factors may contribute to the longer delay time for force-induced ER calcium release.

Again we want to emphasize that our model does not rule out biochemical signal-mediated mechanisms. As a matter of fact, biochemical signal-mediated mechanisms, such as protein-protein interaction and cytoskeletal remodeling under mechanical tension are important to mediate mechanical signals as discussed above. We have also incorporated the discussion of this time delay of ER calcium release in the revised manuscript (please see the eighth paragraph of the Results and Discussion section).